**Data Availability Statement:** The data used to support the results of this study is available from open databases without restriction. 1. The dataset of disease and affected countries was arranged due to the following websites of WHO, readers can aslo

# Trade shocks and trade diversion due to epidemic diseases: Evidence from 110 countries

**Naixi Liu**[1☯], **Yu Li**[2☯], **Mingzhe Jiang**[1☯], **Bangfan Liu**[3,4]*

1 School of International Economics, China Foreign Affairs University, Beijing, China, 2 College of Economics and Management, China Agricultural University, Beijing, China, 3 School of Public Administration Yanshan University, Qinhuangdao, China, 4 Hebei Public Policy Evaluation and Research Center, Qinhuangdao, China

☯ These authors contributed equally to this work.

* liubangfan@ysu.edu.cn

## Abstract

COVID-19 has been a massive trade shock that has disrupted global trade, making the last few years a special phase. Even during normal times, epidemic diseases have acted as trade shocks in specific countries, albeit not to the same extent as COVID-19. For some trade shocks, the situation normalizes after the disease transmission is over; for some, it does not. Thus, specific countries can sometimes lose their original trade ratio due to trade diversion; that is, an epidemic disease could lead to unexpected industry restructuring. To examine this, based on data on 110 WHO members from 1996 to 2018, we use a fixed-effect panel model supported by the Hausman Test to empirically identify whether epidemic diseases can cause trade shocks and trade diversion. We find: First, epidemic disease can lead to negative shocks to a country's trade growth and its ratio of worldwide trade. Second, with a longer epidemic, the probability of the trade diversion effect increases. Our results hold even after considering country heterogeneity. This presents a considerable concern about the shock of COVID-19 lasting further. Many countries are not just facing the problem of temporary trade shocks, but also the challenge of trade diversions. In particular, the probability of trade diversions is increasing rapidly, especially for late-developed countries due to their lack of epidemic containment and vaccine-producing capabilities. Even middle and high income countries cannot ignore global industry chain restructuring. Forward-looking policies should be implemented in advance; it may be too late when long-term trade damage is shown.

## 1. Introduction

Since the COVID-19 outbreak, the global economy has suffered from massive shocks due to the impact of the pandemic as well as other reasons. As threats are decreasing, the economy is recovering. Crucially, one may ask whether the trade shock due to COVID-19 will disappear or leave a huge scar. While COVID-19 had global and wide ranging impacts, the World Health

get the access to the public data used in the paper due to sent email to them. https://www.who.int/emergencies/disease-outbreak-news https://www.who.int/csr/don/archive/disease/en/ https://www.who.int/csr/don/archive/country/en/ https://www.who.int/emergencies/diseases/en/ 2. The dataset of trade was arranged due to the following link of WTO: https://data.wto.org/en 3. The dataset of Macroeconomic Variables was arranged based on the following link of WORLDBANK: https://data.worldbank.org/indicator.

**Funding:** This research was funded by China Foreign Affairs University Thinktank Project "Globalization and Regionalization: A Study on the Law and Trend of the Development of International Economic and Trade Organizations" (Project No. 3162021ZK01),and by Hebei Provincial Department of education science research plan project major research project of Humanities and Social Sciences "Research on the promotion path of industrial chain of Hebei coastal counties (districts and cities) from the perspective of coordinated development of Beijing, Tianjin and Hebei" (Grant No. ZD202104) ", and by the Annual project of Hebei Social Science Foundation "Research on the development history of Hebei coastal cities" (No.: HB18WH06). The funders had no role in study design, data collection and analysis, decision to publish, or preparation of the manuscript.

**Competing interests:** We declare that we do not have any commercial or associative interest that represents a conflict of interest in connection with the work submitted.

Organization's (WHO) data show that regional epidemic diseases always exist somewhere in the world and produce trade shocks in specific counties even during normal times. Sometimes, the diseases have impacts for a very short time or limited scale; hence, the trade shock is just a temporary interruption and things recover after the transmission of the disease decreases [1]. However, sometimes, trade partners can lose patience due to long-term supply chain interruptions; or employees and workers can lose confidence about returning to their jobs, which may force them to find new work [2, 3]. Furthermore, the plants and machines will depreciate due to extensive idle time. Together, these negative impacts can mean that the impacts of some trade shocks can be un-recoverable [4, 5]. Countries suffering from epidemic diseases can also face unexpected industry restructuring due to trade shock and trade diversion [6, 7], which can have adverse exogenous impacts on their economies [8–10].

Then, some crucial questions naturally arise: To what extent do epidemic diseases lead to trade shocks and trade diversions? Will COVID-19 lead to worldwide industry restructuring due to its widespread impact? We argue that the prior literature has two limitations which prevents it from answering these two important questions.

First, few empirical studies examine the impact of epidemics on trade. Some research does exist, such as that on the temporary stagnation of trade in the short term [11–13]. However, most only undertake qualitative discussions of trade diversion [14, 15]. Second, few empirical studies investigate the trade diversion caused by epidemics. Some scholars have discussed trade shocks and trade diversions [12–16], with few on the latter, especially empirical ones.

We innovatively use balanced panel data of the disease records of 110 member countries released by the WHO from 1996 to 2018, and their macro-data from the World Bank and WTO to empirically examine whether epidemic diseases lead to trade shock and trade diversion. We do not use the data after and including 2019 because COVID-19 is very different from the regional and limited epidemic diseases in the past. COVID-19 impacted the global economy because it was so widespread, which made it very special event even in human history [17–22]. It also made the global macro-data including and after 2019 a substantially inter-related set that cannot be identified by cross-sectional estimation. Hence, to achieve more precise identification, we only retained the data before 2019. Nevertheless, our conclusions provide valuable insights for understanding the influence of COVID-19. Finally, we perform estimations considering country heterogeneity to test validity and robustness of our baseline model.

In essence, this study examines the impact of infectious diseases on economic development, which is the field of "infectious disease economics." Epidemiological economics is a developing discipline, and its theory is neither mature nor systematic. We find only few works that have directly studied the economics of infectious diseases. In the WOS, only a dozen papers are directly on infectious diseases and economics, with only two directly discussing the relationship between infectious diseases and economic development. Nicola points out that epidemics, infections, and diseases affect a large number of individuals across developing and developed countries [23]. Gong Mengchun et al. stated that in communicable disease epidemics, early initiation of epidemiological and clinical data collection and analysis, and conducting relevant research are essential for successful epidemic containment. The authors propose the evaluation methods for case reports, randomized controlled trials (RCTs), real-world evidence studies, and health economics research during an epidemic. Data from health economics research also provide important support for optimizing communicable disease prevention and control strategies. Herein, we summarize the health economics research methods, study designs, and technical aspects during the outbreaks. We recommend incorporating clinical and health economics research into the prevention and control plans, and that measures be taken to ensure both the standards and feasibility of these studies to improve the response capacity

against communicable disease outbreaks [24]. Boucekkine, Raouf Carvajal, etc., [25] and Sung Janet [26] also discussed the relationship between epidemics and economic development. Only 28 works are present on the topic of epidemiological economics in the China National Knowledge Network, among which 4 are directly titled "epidemiological economics": Progress in research on economic impact of pandemics and intervention policies [27], Advances in theoretical and empirical research on epidemiological economics [28], Epidemiological economics and the transformation of health policy [29], and Pandemic economics: the 1918 pandemic influenza and its modern significance [30]. The study of the impact of epidemics on trade shocks and trade diversion is undoubtedly a useful supplement to epidemiological economics.

The remainder of this work is as follows: Section 2 presents the literature review. Section 3 outlines our theoretical framework, where we elaborate on the impact mechanism and path of epidemic diseases on trade shock and trade diversion; further, we outline our hypotheses. Section 4 presents the methodology, including the model construction, data selection, and variable interpretation. Section 5 outlines the estimation results and undertakes the discussion. Section 6 undertakes further structural comparative estimation and robustness test based on country heterogeneity. Finally, Section 7 presents the conclusions of this study.

## 2. Literature review

### 2.1 The impact of epidemics on trade shock and trade diversion

The WHO's "Building a Secure Future: Global Public Health Security in the 21st Century" report notes that disease outbreaks can cause significant economic losses by disrupting trade and travel. Some scholars have also studied the impact of epidemics on trade.

Huang Zhaoyin [14] analyzed the impact of SARS on China's trade and found that trade fluctuations were small during the month of the SARS outbreak. However, because of the lag, business activities were postponed or even cancelled in the following months. Further, the export threshold of Chinese products was raised, export share decreased, and foreign trade was difficult. From the perspective of importing countries, Huang Yueheng et al. [11] linked the setting of technical trade barriers for different countries with the probability of epidemic occurrence in exporting countries; raised the technical import threshold for countries with a high probability of epidemic occurrence to reduce the risk of epidemic impact on domestic trade. Leadon and Herholz [12] presented historical evidence about the introduction of pathogens into disease-free areas by the horse product trade and demonstrated that the disease caused short-term stagnation in horse trade. Shi Changhua and Gong Hanying [13] reviewed various international public health emergencies, and found that health control measures taken by various countries for personnel, entry-exit goods, and vehicles resulted in huge losses to various economies, especially foreign trade. Perrings et al. [16] believed that the emergence and spread of infectious diseases are closely related to the growth of trade networks. When an epidemic occurs, it can quickly spread in all directions of the trade network through trade activities, making importers the driving force in determining trade flow; crucially, imports from the affected countries may be avoided. Shao Bai et al. [15] analyzed the impact of international public health emergencies on countries' trade. The authors found the interruption of foreign economic exchanges and the reduction or delayed transfer of foreign trade orders in the affected countries. Furthermore, countries neighboring the affected countries see a significant shift in trade, resulting in a substantial increase in export demand.

### 2.2 Other impacts on trade and trade diversion

**2.2.1 Trade agreements and trade barriers.** Clausing [31] found that a free trade agreement (FTA) between two countries has a trade creation effect but no trade diversion effect.

Bohara et al. [32] validated the view that "trade integration may exacerbate inefficiency," finding that trade is indeed transferred to free trade area partner countries, but this trade diversion effect is internalized and weakened due to the disappearance of trade barriers. Guo Yan and Li Xiubin [33] studied the zero-quota and zero-tariff textile trade agreements signed by the United States with some developing countries based on the North American Free Trade Agreement (NAFTA). The authors observed that the traditional textile-exporting countries in Asia suffered losses from trade diversion. Zhang Bin and Zhang Shu [34] analyzed the trade diversion of various commodities in the United States under the influence of NAFTA from 1994 to 2003. The authors found an annual discontinuous trade diversion effect on labor-intensive manufactured goods. Magee [35] estimated that regional trade agreements (RTA) have a significant predictive signal effect on trade flows, which will continue to affect trade for the next 11 years. Shen Hao and Yang Yong [36] analyzed the trade diversion effect of African economic integration, finding that the intra-community trade scale was too small to change the trade flow during 1999–2005. Datta and Kouliavtsev [37] studied the effects of wages, tariffs, and exchange rates on the import structure of American textiles before and after the establishment of NAFTA. The authors found that the trade creation effect was significant, while the trade diversion effect was not. Sun and Reed [38] conducted OLS estimations on the trade gravity model and found a trade diversion effect among the FTA members, with a substantial increase in agricultural trade. Baylis and Perloff [39] found that when voluntary price constraints were strong, Mexico shifted its tomato exports from the United States to Canada, whereas Canada increased its tomato product exports to the United States. Cao Liang et al. [40] calculated the trade flow of the China-ASEAN FTA by combining the trade gravity model and demonstrated no import trade diversion but only trade creation. Yang and Martinez [41] also found that the trade agreement has led to huge trade creation in the ASEAN region. Dai et al. [42] collated historical data and concluded that FTAs transfer trade from nonmember countries to member countries. Cheong et al. [43] explored the impact of preferential trade agreements on trade, and observed that they could weaken trade creation and shield the trade diversion effect of future trade agreements, thus minimizing interregional trade losses. Magee [44] estimated the general equilibrium effect of the European Community and found that trade creation generated by the customs union was more than twice that of trade diversion. Based on trade gravity and import demand models, Xu Fen and Liu Hongman [45] found no trade diversion effect in China's agricultural exports to different free trade zones.

**2.2.2 International trade disputes.** Brenton [46] summarized the anti-dumping actions of various countries against the EU and observed that an anti-dumping policy would lead to the transfer of the importing trade of the suing country to non-EU suppliers. Chen Hanlin [47] found that anti-dumping actions initiated by the United States against China greatly reduced China's exports to the United States and that part of the reduction may have been transferred to the EU. Xiang Hongjin [48] found that exports of similar products from South Korea, India, and other countries to the United States increased during the same period. Liu Gravi and Shao Min [49] focused on Indian anti-dumping, listed the differences in Chinese product exports to India, and concluded that the transfer effect of Chinese product trade with comparative advantage was small. Park [50] focused on China's anti-dumping investigation using a panel model and showed that China's anti-dumping protection has significant trade inhibition and trade diversion effects. Ming and Shihui [51] examined international counter-vailing cases from 1993 to 2007 and estimated that countervailing complaints caused the transfer of product imports from the accused country to the non-accused country. Liu Graviti and Cao Jie [52] considered the product categories of EU anti-dumping against China and found that the trade diversion effect was inconsistent with the intensity of anti-dumping; these differences came from product competitiveness and product dependence. Zhang Yong [53] used

anti-dumping cases initiated by the United States from 1994 to 2008 to demonstrate that complaints by it can only increase the import market share of a non-respondent country without trade diversion. Hao Liang [54] constructed an anti-dumping duty model and analyzed the optimal anti-dumping duties imposed by manufacturers simultaneously or successively from the perspective of microeconomics. The author found no significant correlation between the taxation level and strength of the trade diversion effect.

**2.2.3 Economic crisis.** Headey [55] observed changes in international grain export volume and export price before the outbreak of the food crisis in 2008. The author found that the surge in grain prices lagged the rise in export volume. Further, the surge in food demand in the international market caused 15 to 27 food price changes worldwide, which then led to a change in a country's food import demand. Liu Hongguang et al. [56] examined the industrial trade diversion structure of major countries and regions before and after the 2008 financial crisis. The authors argued that China was the largest net industrial transfer country, while the EU, the United States, and other countries gradually became destinations of service trade diversion.

## 2.3 Summary

In summary, the conclusion of trade agreements (or changes in trade barriers), international trade disputes, and economic crises can affect trade, whereas the conclusion of trade agreements and economic crises can lead to medium- and long-term trade diversions. However, few empirical studies examine whether epidemics can impact trade and cause long-term trade diversions. Only a few scholars have conducted qualitative analyses of trade shocks and trade diversions caused by specific epidemics, but no general conclusions can be drawn on this issue. This study provides a supplementary answer to this research question.

## 3. Theoretical framework and hypotheses

We deconstruct and analyze the impact mechanism of epidemics on trade shock and trade diversion. Here, we establish a theoretical framework and propose two empirical hypotheses. First, for trade without intra-industry or intra-multinational trade, there can be three impacts (see Fig 1):

Path ①: During an epidemic disease outbreak, enterprises will spontaneously restrict production or stop work due to the panic and collective action to prevent and control the spread of epidemic disease. Then, foreign suppliers and downstream manufacturers will be forced to divert their business to enterprises in other countries. The trade chains of the affected countries will be interrupted, impacting foreign trade and creating short-term trade diversions. However, as the epidemic wanes, enterprises in the affected countries tend to reduce their costs due to overcapacity [57]. Furthermore, the country's products and services will recover a certain degree of international competitiveness, thus slowing down the original trade diversion. However, owing to shrinking demand during the epidemic, when production resumption causes overcapacity, actually resuming production is difficult. Then, enterprises may not recover their original international market share, resulting in long-term trade diversion.

Path ②: An epidemic outbreak also causes "safety concerns" in the international market for products and services exported by affected countries [58, 59]. For example, import restrictions or higher import standards are imposed on products and services exported by the affected countries, which affects their export trade. For example, during COVID-19, in 2020, South Korea, Indonesia, and other countries restricted agricultural imports from China. Meanwhile, the original trading partners turn to trade with "safer countries", and the affected countries are forced to suffer from trade diversion. Moreover, to control the spread of the epidemic,

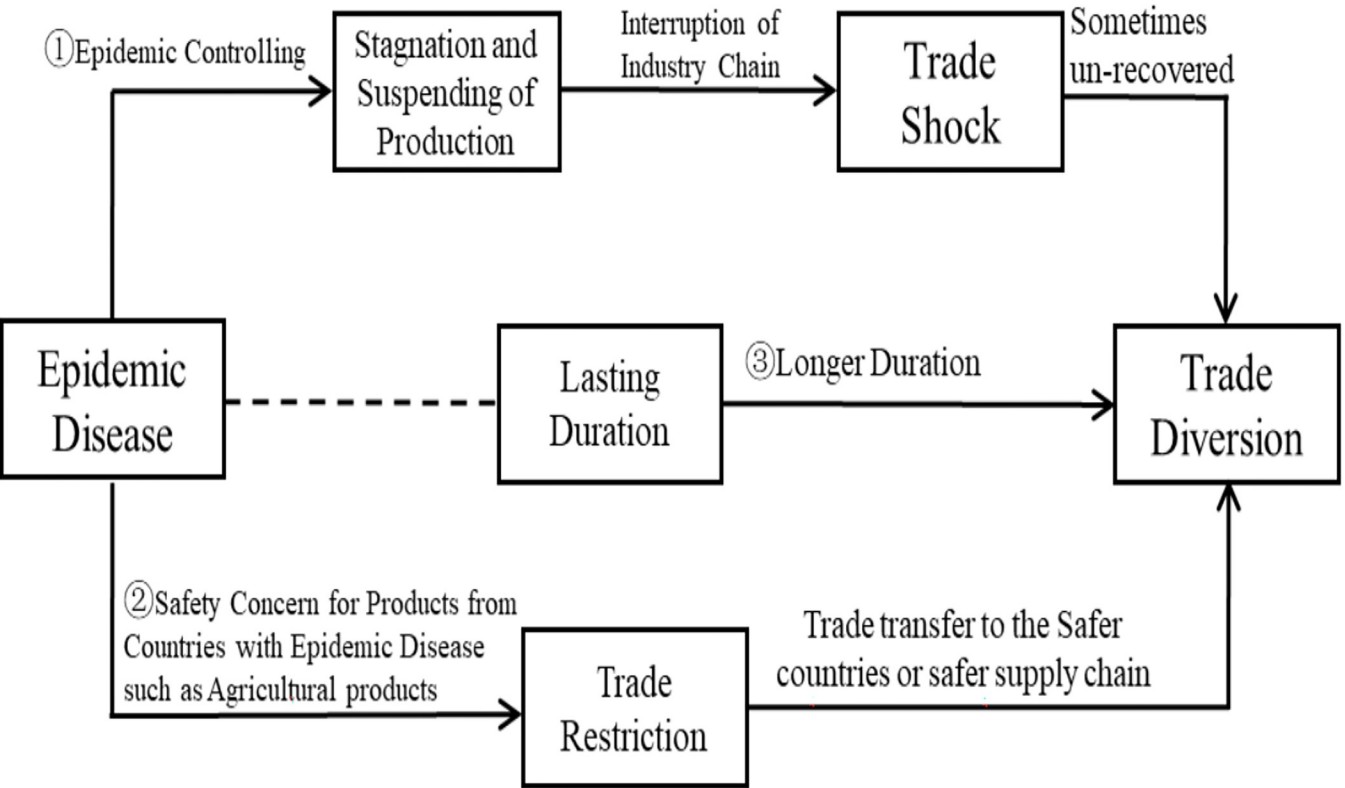

**Fig 1. Logical diagram of the impact of the disease on trade.** Note: Arrows indicate the transmission mechanism of the impact of epidemic diseases on trade shocks and diversion.

commercial travel with the affected countries may also be restricted [60]; this can worsen the decrease in trade.

Path ③: With the epidemic stretches longer and more countries are affected, it is easy to form a rotation to delay the spread of the epidemic. Then, the impact on trade increases and the probability of a trade shock evolving into trade diversion also increases.

Second, when the global industry is not in recession, replacing intra-industry and intra-multinational trade is more challenging. However, when an epidemic occurs, trade shocks and temporary trade diversion occur. Yet, the possibility of long-term trade diversion is relatively low after the epidemic ends. However, when the global industry falls into a recession, the intra-industry trade remains low for a long time. This can impact the original trade chain and eventually lead to long-term trade diversion.

Based on this discussion, we propose two hypotheses:

Hypothesis 1: Epidemic diseases will lead to trade shock.

Hypothesis 2: Epidemic diseases will lead to trade diversion.

## 4. Methodology

### 4.1 Data selection

We use disease data collected from the WHO website releases and news reports, and macro-economic data from the World Bank and WTO databases. After excluding countries with serious data shortages, a balanced panel dataset of 110 countries during 1996–2018 was obtained.

We could not include samples after 2019 and COVID-19 because of two challenges. First, COVID-19 is so different from all epidemics during the last two decades given its global impact [17–22]. Hence, macroeconomic variables may be substantially interrelated such that the cross-sectional or panel identification during these three years may be difficult. Our focus is on the extent to which endogenous impacts, such as epidemics, could influence trade and reshape the industrial structure via trade shock and trade diversion. These can only be identified when the total trade is still on the normal growth path while some trade disappears. During the COVID-19 pandemic, global trade decreased throughout the world, increasing the difficulty of identifying trade shocks or diversions for specific countries. Second, during the last three years, the entire world has been fighting COVID-19, let alone the WHO. The usual regional epidemic have been shadowed by COVID-19, and the WHO does not have a massive team to update and trace information on all epidemics. Further, government statistics were focused on COVID-19 and chose to ignore the original epidemics. Considering these two practical restrictions, to achieve more precise identification, we chose samples during 1996–2018.

## 4.2 Model construction

Despite the above-mentioned limitations, we still need a suitable method for identifying trade shocks and trade diversion. In past research, especially in dynamic research topics, most scholars used the first-order form of one variable to identify the shock or interruption of its normal development, and used a ratio set of one variable to illustrate the structural change of the system to which the variable belongs. This provides an easy way to capture the impact of epidemic diseases on trade. We use the growth rate of trade variables to identify whether the trade shock has occurred with the spread of the epidemic disease. Further, we use the trade variables' share in the world to identify whether trade diversion has occurred during the epidemic. Based on the two proposed hypotheses and referring to the general panel estimation model setting, we set our model as follows:

Eqs I and II estimate the impact of a trade shock on export and overall trade growth:

$$
ExportGrowth_{it} = \frac{Export_{it}}{Export_{i,t-1}} = \beta_0 + \beta_1 EDI_{it} + \beta_2 NEDI_{it} + \beta_3 EDITime_{it} + \beta_4 EDITime_{it}^2
$$
$$
+\beta_5 NEDITime_{it} + \beta_6 NEDITime_{it}^2 + \beta_7 RTAIn_{it} + \beta_8 Disputes_{it} + \sum_{k=9}^{m} \beta_k Z_{it} + \eta_{it} \tag{I}
$$

$$
TradeGrowth_{it} = \frac{Trade_{it}}{Trade_{i,t-1}} = \beta_0 + \beta_1 EDI_{it} + \beta_2 NEDI_{it} + \beta_3 EDITime_{it} + \beta_4 EDITime_{it}^2
$$
$$
+\beta_5 NEDITime_{it} + \beta_6 NEDITime_{it}^2 + \beta_7 RTAIn_{it} + \beta_8 Disputes_{it} + \sum_{k=9}^{m} \beta_k Z_{it} + \gamma_{it} \tag{II}
$$

Eqs III and IV estimate the diversion impact on export and overall trade in the world:

$$
ExportRatio_{it} = \frac{Export_{it}}{Export_{wt}} = \beta_0 + \beta_1 EDI_{it} + \beta_2 NEDI_{it} + \beta_3 EDITime_{it} + \beta_4 EDITime_{it}^2
$$
$$
+\beta_5 NEDITime_{it} + \beta_6 NEDITime_{it}^2 + \beta_7 RTAIn_{it} + \sum_{k=8}^{m} \beta_k Z_{it} + \nu_{it} \tag{III}
$$

$$
TradeRatio_{it} = \frac{Trade_{it}}{Trade_{wt}} = \beta_0 + \beta_1 EDI_{it} + \beta_2 NEDI_{it} + \beta_3 EDITime_{it} + \beta_4 EDITime_{it}^2
$$
$$
+\beta_5 NEDITime_{it} + \beta_6 NEDITime_{it}^2 + \beta_7 RTAIn_{it} + \sum_{k=8}^{m} \beta_k Z_{it} + \tau_{it} \tag{IV}
$$

where i represents the country, t represents the corresponding year, and w represents the world level.

This model is constructed based on two considerations: First, according to hypothesis H1, we use the trade growth rate of a country in a certain year as the dependent variable in Models I and II to estimate the impact of the epidemic disease on a country's trade. This is because trade growth tends to have a certain inertia, and the impact of epidemics is generally reflected in its impact on its original growth trend rather than the absolute value.

Second, according to hypothesis H2, we use the ratio of a country's trade volume to the world's total trade volume in a certain year as the dependent variable in Models III and IV to estimate the trade diversion impact. This is because, in the very short term, such as in the nearest two years, the world trade structure may not undergo sudden changes. Thus, in an equilibrium without exogenous shocks, such as a trade diversion impact, the trade volume of a country should keep growing while the total world trade volume increases. The inconsistency between the growth rates of the trade volumes of a country and the world may be because the country's original trade was replaced by trade from other countries, or the country failed to enter the new expanding market. We can reasonably believe that a change in the ratio of a country's total trade volume to the world's total trade volume can reflect a change in the country's trade structure.

## 4.3 Variables

**4.3.1 Dependent variables.**   Export Growth$_{it}$ and Export Ratio$_{it}$ represents, respectively, the growth rate of exports, and the ratio of exports to the world's total exports$_{it}$ (the value of a country's trade, unit: USD million) and total world exports Export$_{wt}$ (world exports, unit: million US dollars), respectively.

Trade Growth$_{it}$ and Trade Ratio$_{it}$, respectively, represent the growth rate of imports and exports, and the ratio of imports and exports Trade$_{it}$ (a country's trade imports and exports, unit: million US dollars) and to the world's total imports and exports trade$_{wt}$ (world imports and exports, unit: millions of US dollars).

**4.3.2 Independent variables.**   EDI$_{it}$: Epidemic Disease Impact represents the impact of epidemic diseases in country i in year t. This variable is calculated as the sum of the case fatality rates of different epidemic diseases multiplied by the square of the affected range (number of affected countries). We use the case fatality rate to reflect the economic impact of epidemic diseases because when there is a probability of death, the social panic of epidemic diseases will intensify. This will lead to spontaneous stagnation of economic activities and induce traders in affected countries to voluntarily reduce their trade activities. The squared affected range is used because when multiple countries are hit by a pandemic, international trade tends to spiral downward and become more skewed towards only "safer countries," exacerbating the trade shock and trade diversion. In the absence of an epidemic, the value of this variable is 0.

NEDI$_{it}$: Non-Epidemic Disease Impact represents the impact of non-epidemic diseases in country i in year t. This variable is calculated from the sum of the values of the affected range (number of affected countries) times a 0–1 dummy variable to express whether there is a non-epidemic disease in a specific country and year. We only weight the number of affected countries without considering the fatality rate because non-epidemic diseases are less contagious and less likely to cause widespread social panic. When there is no non-epidemic disease, the value of this variable is 0.

EDITime$_{it}$: It represents the duration of epidemic disease. The value of the variable in the year of epidemic outbreak is 1, and increases by 1 for each additional year thereafter. As society and the government will try their best to prevent the disease from spreading and people will

get used to the disease threats, the duration of the outbreak and trade diversion effect may have a non-linear relationship. Hence, both the first and quadratic forms of EDITimeit are added to the estimation equation.

NEDITimeit: It represents the duration of a non-epidemic disease. The value of the variable in the year of the outbreak of a non-epidemic disease is 1 and increases by 1 for each additional year thereafter. Similar to EDITime, both the first and quadratic forms of NEDITimeit are added to the estimation equation.

**4.3.3 Control variables.** RTAInit: Regional Trade Agreements Increase (Regional Trade Agreements Increase) is defined as an increase in the number of RTAs signed by country i in year t. The data are from the WTO Trade Agreement database. This variable is used to control for the possible impact of new trade agreements signed by countries.

Disputesit: It is the number of cases in which country i was the litigant in international trade disputes in year t. The number of trade disputes in all years between the time a dispute was raised and its settlement was reached is added to the value of this variable. Considering that the impact of international trade disputes on a country's trade is mainly manifested as the impact on the change in trade growth rate in the current year [47, 48, 51], there is no inevitable causal logical connection with trade diversion [53, 54]. Therefore, we only added international trade dispute control variable to Models I and II.

Zit: It is a group of control variables added by referring to previous research to mainly control for other macroeconomic variables affecting trade changes, as shown in Table 1.

**4.3.4 Endogeneity problem.** For more effective estimation, we need to consider whether epidemics are endogenous variables that affect a country's trade growth rate and trade ratio. As a natural disaster that is difficult to predict in advance, an epidemic is more likely to be seen as an endogenous variable in this study. The spread of an epidemics is often due to the cross-border and cross-regional movement of people, but not trade flow [61]. Therefore, the possible endogeneity of epidemics can be reasonably ignored.

# 5. Estimation results and discussion

## 5.1 Descriptive statistics

The descriptive statistics of variables are shown in Table 2.

**Table 1. Control variables—macroeconomic variables.**

| Variable | Meaning | Variable description |
|---|---|---|
| *GDPGrowth* | GDP growth rate | Reflecting national income and national economic development. The GDP growth and trade growth of a country have an endogenous relationship. |
| *POPGrowth* | Population growth rate | Population growth drives demand growth, which in turn boosts imports |
| *MGrowth* | Broad money growth rate | Currency increase can reduce short-term nominal interest rate, but also cause a country's currency relative depreciation, expand export trade, and reduce import trade; and vice versa. |
| *R* | Effective interest rate | The rise in the real interest rate leads to capital inflow and relative currency appreciation, which expands import trade and reduces export trade; meanwhile, the rise in savings under liquidity preference will expand trade surplus and vice versa |
| *Agriculture* | Agricultural value added as a percentage of GDP | Reflecting the structure of the primary industry and influencing the export structure of agricultural products; it is particularly important for agricultural powers |
| *Industry* | Industrial value added as a percentage of GDP | This reflects the structure of the secondary industry and affects the export structure of industrial products; it is is particularly important for industrial powers |

**Table 2. Descriptive statistics of key variables.**

| Variable | Mean | Standard deviation | Minimum value | Maximum value |
|---|---|---|---|---|
| *ExportGrowth* | 1.0923 | 0.2392 | 0.2941 | 4.7143 |
| *TradeGrowth* | 1.0842 | 0.1664 | 0.3581 | 1.9407 |
| *ExportRatio* | 0.0048 | 0.0144 | 3.50 e-07 | 0.1375 |
| *TradeRatio* | 0.0049 | 0.0156 | 1.85 e-06 | 0.1549 |
| *EDI* | 363.1312 | 1230.729 | 0 | 5854.32 |
| *NEDI* | 0.2217 | 1.3007 | 0 | 18 |
| *EDITime* | 0.4043 | 1.1682 | 0 | 14 |
| *NEDITime* | 0.0806 | 0.3894 | 0 | 7 |
| *RTAIn* | 0.3162 | 1.0346 | 0 | 28 |
| *Disputes* | 0.5711 | 2.1784 | 0 | 28 |
| *GDPGrowth* | 3.9657 | 4.2499 | 30.1451 | 88.9577 |
| *POPGrowth* | 1.5142 | 1.3285 | 9.0806 | 8.1179 |
| *MGrowth* | 17.4821 | 85.2695 | 99.8637 | 4105.573 |
| *R* | 7.3454 | 10.6454 | 93.5135 | 130.345 |
| *Agriculture* | 14.1097 | 12.3015 | 0.0249 | 79.0424 |
| *Industry* | 26.2827 | 11.2967 | 3.1502 | 74.1130 |

## 5.2 Estimation results and discussion

Due to significant country heterogeneity and macroeconomic cycle effects, country- and year-fixed effects should be controlled. However, considering the possible partial sampling effect of the sample data, we still need to test its measurement characteristics. The fixed and random effects models were used for the estimation, and then the Hausmann test was used to select the final model. The results show that (Table 3) selecting a fixed-effects model is more appropriate.

The macroeconomic variables used in our estimation all have first-order derivative properties (the discrete form is reflected in the growth rate or ratio). Since the number of estimated cross-section samples is much larger than the number of years, a separate unit root test is not required [62, 63]. Therefore, we estimate the fixed effects models of Eqs (I) to (IV), respectively, and the estimated results are shown in Table 4. We find: First, Epidemic EDI has a significant negative impact on both export and trade growth of a country's export trade. This supports our empirical hypothesis. The non-epidemic epidemic disease NEDI has no significant impact on the export and trade growth; thus, non-epidemic epidemic disease has no significant impact on a country's trade.

Second, the EDI of an epidemic disease has a significant negative impact on both ExportRatio and TradeRatio. Thus, an epidemic disease has a long-term trade diversion effect. The influence of NEDI is not significant, indicating no long-term trade diversion effect.

**Table 3. Hausmann test results.**

| Model | I | II | III | IV |
|---|---|---|---|---|
| $\chi^2$ value | 91.18*** | 124.96*** | 143.55*** | 145.87*** |
| P values | 0.000 | 0.000 | 0.000 | 0.000 |

Note:

* $p < 0.05$

**$p < 0.01$

***$p < 0.001$.

**Table 4. Estimation results.**

|  | Model I | Model II | Model III | Model IV |
|---|---|---|---|---|
|  | *ExportGrowth* | *TradeGrowth* | *ExportRatio* | *TradeRatio* |
| *EDI* | -0.0000125** | -0.0000150*** | -0.000000444*** | -0.000000428*** |
|  | (-2.82) | (-5.18) | (-5.26) | (-5.52) |
| *NEDI* | 0.00635 | 0.00570 | 0.000147 | 0.0000820 |
|  | (1.40) | (1.92) | (1.70) | (1.03) |
| *EDITime* | 0.0261** | 0.0270*** | 0.00138*** | 0.00126*** |
|  | (2.68) | (4.22) | (7.41) | (7.37) |
| *EDITime$^2$* | -0.00282* | -0.00300*** | -0.0000780*** | -0.0000730*** |
|  | (-2.41) | (-3.91) | (-3.49) | (-3.55) |
| *NEDITime* | -0.0173 | -0.0318 | -0.00199*** | -0.00173*** |
|  | (-0.63) | (-1.78) | (-3.82) | (-3.61) |
| *NEDITime$^2$* | 0.00721 | 0.0105** | 0.000225 | 0.000169 |
|  | (1.18) | (2.61) | (1.93) | (1.58) |
| *RTAIn* | 0.00738 | 0.00759* | 0.000161 | 0.000178* |
|  | (1.59) | (2.49) | (1.87) | (2.25) |
| *Disputes* | -0.00210 | -0.00374 | - | - |
|  | (-0.58) | (-1.57) | - | - |
| *GDPGrowth* | 0.0214*** | 0.0181*** | 0.000000637 | 0.00000305 |
|  | (18.81) | (24.22) | (0.03) | (0.15) |
| *POPGrowth* | -0.00655 | -0.0102* | -0.0000474 | -0.0000401 |
|  | (-0.96) | (-2.26) | (-0.36) | (-0.33) |
| *MGrowth* | -0.000103 | -0.0000740* | -0.000000453 | -0.000000280 |
|  | (-1.86) | (-2.04) | (-0.43) | (-0.29) |
| *R* | -0.00530*** | -0.00436*** | -0.000000844 | 0.000000286 |
|  | (-9.85) | (-12.34) | (-0.08) | (0.03) |
| *Agriculture* | 0.000116 | -0.000209 | -0.0000961*** | -0.0000958*** |
|  | (0.10) | (-0.27) | (-4.25) | (-4.61) |
| *Industry* | 0.00302* | 0.00350*** | -0.0000206 | -0.0000254 |
|  | (2.43) | (4.30) | (-0.87) | (-1.17) |
| **Term of constant** | 0.973*** | 0.971*** | 0.00651*** | 0.00679*** |
|  | (22.91) | (34.85) | (8.06) | (9.14) |
| **Number of samples** | 2530 | 2530 | 2530 | 2530 |

Note:

* $p < 0.05$

** $p < 0.01$

*** $p < 0.001$; values in parentheses are t-values.

Third, the regression coefficient of epidemic duration EDITime shows that the original (quadratic) form has a positive (negative) impact on trade growth and trade ratio. This indicates that the negative impact of epidemic duration on trade has a concave feature. That is, with the initial stabilization of the epidemic spread and improvement in market expectations, especially the rapid economic recovery after a short-term epidemic, the impact of the epidemic on trade is alleviated. Meanwhile, the demand for epidemic control facilities and health materials results in a small increase in short-term trade. However, with a further increase in the duration of the epidemic, the confidence to overcome the epidemic becomes unstable. As the epidemic spreads to more countries, the external environment deteriorates, short-term trade

substitution gradually transitions to long-term trade diversion, and the impact of the epidemic on trade strengthens. NEDI has a significant negative impact on the trade ratio. This may be because non-epidemic diseases have a small range of infections and cannot affect the external environment. Meanwhile, the short-term decline in their own trade decreases the ratio. Due to the relatively short duration of non-epidemic diseases, secondary forms do not show significant effects.

Fourth, RTAIn has a significant positive effect on trade growth, indicating that more RTAs have a trade creation effect. RTAIn also has a significant positive impact on the trade ratio. This indicates that the trade creation effect due to the increase in RTAs is stronger than the trade substitution and crowding-out effects. This finding is consistent with most previous studies.

Fifth, the GDP growth, monetary growth, and effective interest rates significantly affect a country's trade growth but not its trade ratio. GDP growth can promote a country's trade growth, consistent with our theoretical expectations. An increase in money growth has a negative inhibitory effect on trade growth because the demand response is faster and more flexible than the supply response. The short-term inhibitory effect of money growth on imports is greater than its short-term promotional effect on exports. An increase in the real interest rate also has a negative inhibitory effect on trade growth. This indicates that the negative impact on trade through capital flows and exchange rates is greater than the positive impact on trade through investment and savings growth.

Sixth, the ratio of agricultural value-added to GDP has a significant negative impact on a country's trade ratio. Generally, the faster the growth in exports of agricultural products (natural-resource-intensive products), the slower the growth in exports of industrial products. In the past two decades, the proportion of agricultural trade in international trade has been declining, while the trade of industrial goods and services has increased. Thus, the proportion of trade of agricultural exporting countries in the world has decreased.

Seventh, the ratio of industrial value added to GDP has a significantly positive impact on a country's trade growth. This indicates that an increase in a country's industrial output value can increase exports of industrial products, thus promoting trade growth. This positive correlation is consistent with general facts especially for countries dominated by industrial exports (labor-intensive).

Multicollinearity does not influence the significance of our estimation above. Furthermore, in the fixed-effect panel estimation process, no variable is omitted; this means that the influence of multicollinearity could be very small. Still, to exclude this influence more precisely, we take the variable EDI and control variables to conduct the variance inflation factor (VIF) test. The results listed in Table 5 show that the probability of multicollinearity is very less. Thus, our estimation results and findings are appropriate.

## 6. Structural analysis and robustness test

To better understand the impact of the epidemic on different types of countries, we conducted further structural analysis and robustness tests.

### 6.1 Comparison of the impact of epidemics on trade between large and small countries

Due to their dominant position in international trade, epidemics are more likely to affect large countries' trade. The complex industrial structures of large countries make it more difficult to recover from the epidemic's impact, which may lead to long-term trade diversion. For small countries, the impact on trade growth is relatively small. Here, we use the 2018 median GDP

**Table 5. VIF test results.**

| Variable | VIF | 1/VIF |
|---|---|---|
| *EDI* | 1.75 | 0.571 |
| *EDITime* | 1.73 | 0.578538 |
| *RTAIn* | 1.08 | 0.928057 |
| *Disputes* | 1.06 | 0.941688 |
| *GDPGrowth* | 1.28 | 0.780812 |
| *POPGrowth* | 1.23 | 0.815188 |
| *MGrowth* | 1.26 | 0.793769 |
| *R* | 1.06 | 0.943815 |
| *Agriculture* | 1.57 | 0.638144 |
| *Industry* | 1.3 | 0.766417 |

of each country published by the World Bank to classify large and small countries, and estimate the total impact of the epidemic on import and export trade. The results are shown in Table 6.

In both groups, an epidemic disease has a significant negative impact on trade growth. Further, it has a significant negative impact on the trade ratio of a large country, consistent with previous results. In the large country groups, the impact of epidemic duration is also consistent with the original estimates. This indicates that for large countries, the impact of an epidemic disease is more likely to lead to long-term trade diversion. Meanwhile, the impact of an epidemic disease mainly manifests as a short-term trade shock for small countries. Non-epidemic diseases positively affect the trade growth of large countries. Thus, these diseases do not have a strong impact on original trade activities, but cause a small increase in trade due to the

**Table 6. Group estimation results: large and small countries.**

|  | Model II Large countries | Model II Small countries | Model IV Large countries | Model IV Small countries |
|---|---|---|---|---|
|  | *TradeGrowth* | *TradeGrowth* | *TradeRatio* | *TradeRatio* |
| *EDI* | -0.0000137*** | -0.0000153* | -0.000000664*** | -3.33e-10 |
|  | (-4.39) | (-2.54) | (-5.12) | (-0.21) |
| *NEDI* | 0.00723* | -0.00487 | 0.0000650 | 0.00000366 |
|  | (2.50) | (-0.57) | (0.54) | (1.62) |
| *EDITime* | 0.0285*** | 0.0127 | 0.00192*** | -0.0000104* |
|  | (4.12) | (0.81) | (6.68) | (-2.53) |
| *EDITime$^2$* | -0.00326*** | 0.000224 | -0.000132*** | 0.000000945 |
|  | (-4.28) | (0.07) | (-4.15) | (1.20) |
| *NEDITime* | -0.0248 | -0.0117 | -0.00218** | -0.0000278 |
|  | (-1.41) | (-0.13) | (-2.97) | (-1.21) |
| *NEDITime$^2$* | 0.00930* | 0.00334 | 0.000218 | 0.0000194 |
|  | (2.46) | (0.06) | (1.38) | (1.42) |
| **Term of constant** | 0.820*** | 1.058*** | 0.0161*** | 0.000120*** |
|  | (18.20) | (28.84) | (8.66) | (12.43) |
| **Number of samples** | 1265 | 1265 | 1265 | 1265 |

Note:

* $p < 0.05$

** $p < 0.01$

*** $p < 0.001$; values in parentheses are t-values.

purchase of related medical supplies (such as imported drugs and influenza vaccines). In addition, the estimation results of other control variables are consistent with the original baseline estimation results, demonstrating the robustness of the baseline estimations.

## 6.2 Comparison of the impact of epidemics on trade between agricultural and non-agricultural countries

Since the speed of an epidemic disease is highly related to the crowdedness of people and social contact activities, industrial production activities, especially the manufacturing industry, are more susceptible to the epidemic than agricultural production activities; thus, the trade impact of the epidemic on non-agricultural countries might be stronger than that on agricultural countries. We attempted to determine whether such structural differences exist by further grouping regressions. We used the ratio of agricultural value added to GDP published by the World Bank in 2018 as a standard to classify agricultural and nonagricultural countries. Countries with a ratio of more than 10% were defined as the agricultural group, with the remaining defined as the non-agricultural group. Then, we grouped the estimates for comparative analysis. The results are shown in Table 7.

In both groups, the epidemic disease has a significant negative impact on trade growth and trade ratio, consistent with previous results. Comparing the absolute value of the regression coefficient, the impact of epidemic diseases on the trade growth of agricultural countries is slightly lower than that for non-agricultural countries.

Non-epidemic diseases have significantly positive impact on the trade ratio of agricultural countries, and significantly positive and significantly negative impacts on the trade growth and trade ratio, respectively, of nonagricultural countries. The effect of the non-epidemic

**Table 7. Group estimation results: Agricultural and non-agricultural countries.**

|  | Model II Agricultural countries | Model II Non-agricultural countries | Model IV Agricultural countries | Model IV Non-agricultural countries |
|---|---|---|---|---|
|  | *TradeGrowth* | *TradeGrowth* | *TradeRatio* | *TradeRatio* |
| **EDI** | -0.0000149** | -0.0000161*** | -0.000000466*** | -0.000000425*** |
|  | (-3.18) | (-3.76) | (-3.67) | (-3.89) |
| **NEDI** | -0.000720 | 0.00909** | 0.000541*** | -0.000224** |
|  | (-0.12) | (2.83) | (3.45) | (-2.74) |
| **EDITime** | 0.0255** | 0.0183 | 0.00137*** | 0.00133** |
|  | (3.20) | (1.05) | (6.37) | (3.00) |
| **EDITime$^2$** | -0.00289** | 0.00176 | -0.0000764** | -0.000235* |
|  | (-3.19) | (0.43) | (-3.12) | (-2.26) |
| **NEDITime** | -0.00468 | -0.0506 | -0.00388*** | 0.00220** |
|  | (-0.17) | (-1.69) | (-5.31) | (2.88) |
| **NEDITime$^2$** | 0.00665 | 0.00996 | 0.000547*** | -0.00122*** |
|  | (1.27) | (0.89) | (3.87) | (-4.31) |
| **Term of constant** | 1.101*** | 0.844*** | 0.00801*** | 0.00599*** |
|  | (24.71) | (24.96) | (6.69) | (6.99) |
| **Number of samples** | 1334 | 1196 | 1334 | 1196 |

Note:

* $p < 0.05$

** $p < 0.01$

*** $p < 0.001$; values in parentheses are t-values.

disease duration on agricultural and non-agricultural countries is also the opposite. This indicates that while epidemic diseases have a greater impact on industrial production in general, non-epidemic diseases also have a stronger impact on industrial production than on agricultural production. Therefore, regardless of epidemic or non-epidemic diseases, non-agricultural countries are more vulnerable to trade impacts than agricultural countries, resulting in a long-term trade diversion effect.

## 6.3 Comparison of the impact of epidemics on trade between industrial and service countries

Similarly, we use the ratios of industrial and service value added to GDP published by the World Bank in 2018 as the standards to measure industrial and service countries, respectively. Specifically, we define those countries with respective ratios greater than 25% as the industrial and service country groups, respectively. Then, we perform the estimations and comparative analyses for each group. The results are presented in Table 8.

Clearly, epidemics significantly and negatively affected trade growth and trade ratio in both the industrial and service groups. The impact of the epidemic duration was consistent with the original estimates. The absolute value of the regression coefficient shows that the impact on the trade of industrial countries is slightly less than that on the trade of service countries. However, industrial countries are more vulnerable to the long-term impact of the epidemic, resulting in a long-term trade diversion effect. Thus, compared with service countries, industrial countries may find it more difficult to recover their trade after the epidemic. In recent years, the proportion of services trade in global trade has increased. Service countries with high-tech, knowledge products, and emerging service industries as the main players have lower substitutability for their dominant industries. Even under a more serious short-term trade contraction,

**Table 8. Group estimation results: Industrial and service countries.**

|  | Model II Industrial countries | Model II Service countries | Model IV Industrial countries | Model IV Service countries |
|---|---|---|---|---|
|  | *TradeGrowth* | *TradeGrowth* | *TradeRatio* | *TradeRatio* |
| EDI | -0.0000130*** | -0.0000152** | -0.000000558*** | -0.000000371*** |
|  | (-3.51) | (-3.17) | (-4.54) | (-4.34) |
| NEDI | 0.00477 | 0.00673 | 0.000299** | -0.000505*** |
|  | (1.38) | (1.21) | (2.60) | (-5.12) |
| EDITime | 0.0309*** | 0.00917 | 0.00190*** | 0.000762** |
|  | (3.87) | (0.68) | (7.18) | (3.16) |
| EDITime$^2$ | -0.00334*** | 0.000390 | -0.000115*** | -0.000107* |
|  | (-3.96) | (0.16) | (-4.10) | (-2.41) |
| NEDITime | 0.00239 | -0.0630 | -0.00325*** | 0.00305*** |
|  | (0.11) | (-1.65) | (-4.44) | (4.48) |
| NEDITime$^2$ | 0.00600 | 0.0118 | 0.000457** | -0.00141*** |
|  | (1.37) | (0.86) | (3.13) | (-5.81) |
| Term of constant | 0.908*** | 1.038*** | 0.0111*** | 0.00259*** |
|  | (22.13) | (26.98) | (8.17) | (3.78) |
| Number of samples | 1311 | 1219 | 1311 | 1219 |

Note:

* $p < 0.05$

** $p < 0.01$

*** $p < 0.001$; values in parentheses are t-values.

no significant industrial outflow happens and the corresponding trade share can still recover after the end of the epidemic. Thus, there may be no long-term trade diversion effect. Meanwhile, countries that mainly export industrial products are often less developed and services trade growth is slow. Even if they can survive a certain short-term trade contraction because of inventory protection, they are more likely to suffer industrial outflows with long-term trade contraction, and thus, may experience long-term trade diversion and lose market share. Hence, when an epidemic occurs, we should pay special attention to the outflow of industrial production, especially the middle- and low-end manufacturing industries; increase investment; enhance technological growth in industrial production; improve the value addition by industrial production in the international value chain; reduce the substitutability of export products; and ensure the stable development of the industrial structure.

Non-epidemic diseases positively (negatively) affect the trade ratio of industrial (service) countries. The effect of the duration of non-pandemic diseases on the trade ratios of industrial and service countries is also the opposite. This suggests that non-pandemic diseases, such as common influenza, do not halt industrial activity but may shrink service activity.

In conclusion, both epidemic and non-epidemic diseases have a greater impact on service countries in the short term. In the medium to long term, the impact of epidemic (non-epidemic) diseases on industrial (service) countries is greater.

## 6.4 Summary

In summary, the baseline estimation results are relatively robust. The analysis of the new grouping structure also reveals the following. The impact of epidemics on the trade of large countries is greater than that of small countries. Further, the impact of epidemics on the trade of large countries is more likely to result in long-term trade diversion. The impact of epidemic diseases on the trade of agricultural countries is less than that on nonagricultural countries, and nonagricultural countries are more likely to experience long-term trade diversion while suffering the impact of epidemic diseases. In the short term, the trade impact of the epidemic on industrial countries is less than that on service countries. In the medium and long term, industrial countries are more likely to have long-term trade diversion than service countries. Therefore, for countries such as China, which is both a large economy and an industrial country, it is much more important to adopt policies in advance to ensure the stability, transformation, and upgrading of the industrial structure in the long run after an epidemic; this can help in avoiding trade diversion, which can substantially harm the industry structure.

## 7. Conclusions

This study empirically examines the impact of epidemic diseases on trade and long-term trade diversion. Our robust estimation results show that an epidemic disease has deep and significant impacts on a country's the industrial structure, which can affect the stability of a country's long-term economic growth. The COVID-19 outbreak in 2019, and its recurrence in the autumn and winter of 2020 prompted countries worldwide to strengthen epidemic prevention and control efforts. However, some countries, except China, have yet to recover their pre-epidemic economic and trade growth in the post-epidemic era. In April 2021, the International Monetary Fund noted in its World Economic Outlook that many countries would not recover to their pre-pandemic GDP growth levels until 2023, indicating that the general impact of COVID-19 on countries' economic and industrial structures may have a long-lasting impact. China's experience of achieving positive economic and trade growth in a short period during 2021 provides a blueprint for other countries' foreign trade recovery. However, the continued impact of the pandemic on the economy and trade would not be so easy to erase; hence, all

countries should try to strengthen international communication and deepen international cooperation. Among these efforts, first, economic diplomacy should be expanded and effectively implemented, advocating free and multilateral trade, opposing trade protectionism, and gradually resuming trade exchanges. Second, we need to accelerate the building of a community with a shared future for mankind, and advocate an international political and economic order featuring mutually beneficial cooperation and win-win development.

This study presents a brief picture and simple empirical identification of whether an endogenous economic shock can impact trade, and lead to a trade shock or trade diversion. As the COVID-19 pandemic subsides, major countries worldwide have relaxed preventions and controls, aiding the recovery of global trade. However, the insights from this study suggest that many countries may find that they have lost their trade partners and share worldwide trade in the next several years. Furthermore, COVID-19 could be very different from sampled epidemics in this study because it had widespread impacts on a global scale, which may very well lasts for a long time. Both these aspects may mean that COVID-19 will a historical impact [47, 48, 51, 53, 54]. Further research should examine how greatly and deeply it will influence the global trade and industrial structure. Overall, this study uses the historical experience to provide a reference to understand and foresee the huge challenges that many countries may face in the future.

## Acknowledgments

This research was supported by China Foreign Affairs University Thinktank Project "Globalization and Regionalization: A Study on the Law and Trend of the Development of International Economic and Trade Organizations" (3162021ZK01),and by Hebei Provincial Department of education science research plan project major research project of Humanities and Social Sciences "Research on the promotion path of industrial chain of Hebei coastal counties (districts and cities) from the perspective of coordinated development of Beijing, Tianjin and Hebei" (ZD202104) ", and by the Annual project of Hebei Social Science Foundation "Research on the development history of Hebei coastal cities" (HB18WH06), and by National Social Science Foundation "Research on the contemporary value of reshaping ancient Chinese science and technology resources based on historical analysis and cultural interpretation"(20BZX034), and by Major Research project of Humanities and Social Sciences Research of Hebei Provincial Department of Education "Research on the Development of Hebei Province's Characteristic Industries toward the Sea Economy" (ZD202406).

## Author Contributions

**Conceptualization:** Naixi Liu, Mingzhe Jiang, Bangfan Liu.

**Data curation:** Naixi Liu.

**Formal analysis:** Naixi Liu.

**Funding acquisition:** Naixi Liu.

**Methodology:** Mingzhe Jiang.

**Project administration:** Bangfan Liu.

**Resources:** Yu Li.

**Software:** Yu Li.

**Supervision:** Bangfan Liu.

**Validation:** Yu Li.

**Writing – original draft:** Naixi Liu, Mingzhe Jiang, Bangfan Liu.

**Writing – review & editing:** Bangfan Liu.

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
