## [Decision Letter · Decision Letter 0]

11 Jan 2023

PONE-D-22-34151Epidemic Disease Caused Trade Shocks and Long-term Trade Diversion: Evidence from 110 Countries in 1996—2018PLOS ONE

Dear Dr. bangfan,

Thank you for submitting your manuscript to PLOS ONE. After careful consideration, we feel that it has merit but does not fully meet PLOS ONE’s publication criteria as it currently stands. Therefore, we invite you to submit a revised version of the manuscript that addresses the points raised during the review process.

We look forward to receiving your revised manuscript.

Kind regards,

Ruwan Jayathilaka, Ph.D.

Academic Editor

PLOS ONE

“no”

Reviewers' comments:

Reviewer's Responses to Questions

**Comments to the Author**

1. Is the manuscript technically sound, and do the data support the conclusions?

Reviewer #1: Yes

Reviewer #2: Yes

2. Has the statistical analysis been performed appropriately and rigorously? 

Reviewer #1: Yes

Reviewer #2: Yes

3. Have the authors made all data underlying the findings in their manuscript fully available?

Reviewer #1: Yes

Reviewer #2: Yes

4. Is the manuscript presented in an intelligible fashion and written in standard English?

Reviewer #1: Yes

Reviewer #2: No

5. Review Comments to the Author

Reviewer #1: 1. Need to rearrange your abstract as follows: Purpose, Methodology and Design , Findings ( findings of your study is not clearly mentioned ) , Practical implications and Originality

2. As we know, the epidemic disease caused significant global trade shocks and long-term trade diversion during covid pandemic from 2020-2022. So, better to consider your study period up to 2022.

Reviewer #2: I have provided my specific comments on the paper itself and uploaded for the authors' attention.

Overall, this paper needs a language edit from a qualified editor. A lot of ambiguity is present throughout the paper, making the reader very uncomfortable reading. Authors need to maintain consistency in the terminology in use. They must be very careful about the exact meaning of some crucial terms used elsewhere.

Authors must refrain from using first-person references throughout the paper. This has to be Chinese Communist Party (CPC) [NOT "Our Party"]

The introduction needs to be more focused on the impact of epidemics on global trade. Arguments on national economies must be supported by evidence (e.g.: with numbers on the impact on trade creation and diversion)

There is confusion elsewhere in the paper that authors use various terminology in similar meaning but some of the terminology are in fact different in meaning. I have specified them inthe paper. Authors must attend to that and rectify the issues.

6. PLOS authors have the option to publish the peer review history of their article (what does this mean?). If published, this will include your full peer review and any attached files.

Reviewer #1: **Yes: **R. M. Kapila Tharanga Rathnayaka

Reviewer #2: No

---

## [Author Response · Author response to Decision Letter 0]

20 Mar 2023

Response to Reviewers

Respected Editors and Reviewers,

Thanks for your patient reviewing and lots of great suggestions which would make this paper much better. Refer to the suggestions, we conduct very careful and detailed revision according to every great suggestion, which could be summarized to seven major revisions as follow:

1. First, we rearrange the abstract and introduction, to focus on the purpose, methodology, findings and practical implications of this paper. We also revise many critical parts especially the theoretical framework, variable illustration and empirical results discussion, so that this contents could be more precise and clear.

2. Second, we modify the diverse terminology, in the new manuscript we only use trade shock to express the short-term impact which can be recovered after epidemics, and only use trade diversion to express the long-term impact which cannot be recovered even after epidemics. We also revise the other problems with diverse terminology, so that the content could be more focused while reading.

3. Third, we update the reference literatures, especially those published in recent three years. Thanks for your suggestion, we found many great literatures could be very good supplements to enhance several assumption and stylized facts discuss in this research.

4. Fourth, we fatherly conduct the VIF test after baseline estimation to explain the reason why Multicollinearity can be ignored in this paper. Also in the panel estimation there was no variable omitted and the effectiveness of the estimation was still available. Actually, due to that the epidemic disease mostly occurred in a sudden way, often made it an endogenous variable without definite relation to the macroeconomic variables. We think the Multicollinearity problem would not harm the results of this research.

5. We recheck the estimation results and updated the negative expressions. Sorry for that the symbol “-” was lost in the last manuscript. We also adjust the form including titles, subtitles, figs, tables, references etc. according to the journal’s requirements.

6. We keep and modify the results discussion of the comparison between the coefficients even the gap is very small as the reviewer’s question. We considered that small gap still might identify even few implication especially refereeing to the usual observing facts of the realities. However, we do agree with the reviewer’s suggestion. Macro data could also be influence by lots of un-control issues. We modified the expression, turn the view into a discussion of possibility, so that there would be better communication with the readers.

7. Final, we revise the English writing of the whole paper due to help from our native speaker friends. We also erase the unnecessary contents and exclude all of the first-person references throughout the paper so that it could achieve the requirements.

Thanks so much that your comprehensive consideration help us to improve this paper a lot. However, the limitation of this research is that the COVID-19 is not included in the samples, and updating the samples to the year of 2022 is also one of suggestion from the reviewers. The reason why we could not include the samples after 2019 and COVID-19 is due to two challenges. First, the COVID-19 is so different from all of the epidemics during the last two decades with its such a worldwide impact range, the macroeconomic variable became greatly interrelated so that the cross-sectional or panel identification during this three years is not available. Meanwhile, this paper aimed to show to what extent the endogenous impact such as epidemics could impact the trade and reshape the industrial structure via trade shock and trade diversion. These can only be identified when the total trade is still in the normal growth path while some trade disappear. During the COVID-19, the global trade decrease jointly, also made the identification of trade shock or diversion for specific countries not available any more. Second, during last three years, the whole world was fighting with the COVID-19, let alone the WHO. The usual regional epidemics data has been covered by the COVID-19 and WHO has no such a huge team to update the tracing of original epidemics, and also governmental statistics has to focus on COVID-19 and chose to ignore the original epidemics when serious rapid spreading occurs, which make the dataset not continuous if involved samples after 2019. Considering this two practical restrictions, we choose the samples during 1996-2018 to complete this research. Even though, we still think the results of this paper could explain the trade shock and diversion impacted by epidemics, and bring some implication for us to understand or foresee what would happen to the global trade and industrial structure in the next several years. It is believed that after a few years, when we look back, there could be many methods to conduct the analysis of COVID-19 such as take the historical phase as a worldwide random natural experiment or find fantastic breakpoints. In this paper, the results shows that even very few fatal threats could leads to panic and harm the original trade activities. With such a serious influence from COVID-19, we believed that the forward-looking policies and enhancing international economic cooperation should be implemented rapidly while the epidemic turns to Omikjon.

We cannot illustrate every detailed modification in this response, please find all of the revisions in the new manuscript. Great thanks to the editors and reviewers! Without your kind consideration, this paper could not be improved so fast. Looking forward to your further consideration and great suggestion!

Best regards,

All authors of this paper

2023-2-4

---

## [Decision Letter · Decision Letter 1]

13 Apr 2023

PONE-D-22-34151R1Trade Shocks and Trade Diversion of Epidemic Diseases: Evidence from 110 CountriesPLOS ONE

Dear Dr. bangfan,

Thank you for submitting your manuscript to PLOS ONE. After careful consideration, we feel that it has merit but does not fully meet PLOS ONE’s publication criteria as it currently stands. Therefore, we invite you to submit a revised version of the manuscript that addresses the points raised during the review process.

A more convincing literature review is needed to clearly point out the state-of-the-art developments. I suggest adding 5 to 10 recent literature and presenting the innovations of this paper.All references need to be adjusted according to the PLOS ONE guidelines. See https://journals.plos.org/plosone/s/submission-guidelines#loc-references  For example: References are listed at the end of the manuscript and numbered in the order that they appear in the text. In the text, cite the reference number in square brackets (e.g., “We used the techniques developed by our colleagues [19] to analyze the data”). PLOS uses the numbered citation (citation-sequence) method and first six authors, et al.==============================

We look forward to receiving your revised manuscript.

Kind regards,

Ruwan Jayathilaka, Ph.D.

Academic Editor

PLOS ONE

Journal Requirements:

Reviewers' comments:

Reviewer's Responses to Questions

**Comments to the Author**

1. If the authors have adequately addressed your comments raised in a previous round of review and you feel that this manuscript is now acceptable for publication, you may indicate that here to bypass the “Comments to the Author” section, enter your conflict of interest statement in the “Confidential to Editor” section, and submit your "Accept" recommendation.

Reviewer #1: All comments have been addressed

Reviewer #2: All comments have been addressed

2. Is the manuscript technically sound, and do the data support the conclusions?

Reviewer #1: Yes

Reviewer #2: Yes

3. Has the statistical analysis been performed appropriately and rigorously? 

Reviewer #1: Yes

Reviewer #2: Yes

4. Have the authors made all data underlying the findings in their manuscript fully available?

Reviewer #1: Yes

Reviewer #2: Yes

5. Is the manuscript presented in an intelligible fashion and written in standard English?

Reviewer #1: Yes

Reviewer #2: Yes

6. Review Comments to the Author

Reviewer #1: A more convincing literature review is needed to clearly point out the state-of-the-art developments. I suggest adding a literature review and presenting the innovations of this paper.

Reviewer #2: Authors have indicated "No" to ethical approval section. Better mention that the type of research does not require ethical clearance.

7. PLOS authors have the option to publish the peer review history of their article (what does this mean?). If published, this will include your full peer review and any attached files.

Reviewer #1: **Yes: **R. M. Kapila Tharanga Rathnayaka

Reviewer #2: **Yes: **Ajantha Kalyanaratne

---

## [Author Response · Author response to Decision Letter 1]

10 May 2023

Dear editor, we have discussed the research innovation of this paper according to your opinions, and added 8 references directly related to the topic of this paper. Please review it. Thank you for your help!

---

## [Editor Report · Decision Letter 2]

24 May 2023

PONE-D-22-34151R2Trade Shocks and Trade Diversion of Epidemic Diseases: Evidence from 110 CountriesPLOS ONE

Dear Dr. Bangfan,

Thank you for submitting your manuscript to PLOS ONE. After careful consideration, we feel that it has merit but does not fully meet PLOS ONE’s publication criteria as it currently stands. Therefore, we invite you to submit a revised version of the manuscript that addresses the points raised during the review process.

Please make sure you follow the PLOS ONE guidelines. References should be listed at the end of the manuscript and numbered in the order they appear in the text. In the text, please cite the reference number in square brackets. For more details, please refer to the following link:

https://journals.plos.org/plosone/s/submission-guidelines#loc-references

We look forward to receiving your revised manuscript.

Kind regards,

Ruwan Jayathilaka, Ph.D.

Academic Editor

PLOS ONE

Journal Requirements:

Additional Editor Comments:

Please make sure you follow the PLOS ONE guidelines. References should be listed at the end of the manuscript and numbered in the order they appear in the text. In the text, please cite the reference number in square brackets. For more details, please refer to the following link:

https://journals.plos.org/plosone/s/submission-guidelines#loc-references

---

## [Author Response · Author response to Decision Letter 2]

30 May 2023

Dear Editor: Hello, we have rearranged and modified the references according to your requirements, please review. Thank you for your help! liu bangfan,2023/5/31

---

## [Decision Letter · Decision Letter 3]

30 Oct 2023

PONE-D-22-34151R3Trade Shocks and Trade Diversion of Epidemic Diseases: Evidence from 110 CountriesPLOS ONE

Dear Dr. Bangfan,

Thank you for submitting your manuscript to PLOS ONE. After careful consideration, we feel that it has merit but does not fully meet PLOS ONE’s publication criteria as it currently stands. Therefore, we invite you to submit a revised version of the manuscript that addresses the points raised during the review process.

Additionally, our editorial team have significant concerns about the grammar, usage, and overall readability of the manuscript. PLOS ONE requires that published manuscripts use language which is 'clear, correct, and unambiguous', see our criteria for publication at https://journals.plos.org/plosone/s/criteria-for-publication#loc-5. We therefore request that you revise the text to fix the grammatical errors and improve the overall readability of the text. We suggest you have a fluent English-language speaker thoroughly copyedit your manuscript for language usage, spelling, and grammar. If you do not know anyone who can do this, you may wish to consider employing a professional scientific editing service. Whilst you may use any professional scientific editing service of your choice, PLOS has partnered with both American Journal Experts (AJE) and Editage to provide discounted services to PLOS authors. Both organizations have experience helping authors meet PLOS guidelines and can provide language editing, translation, manuscript formatting, and figure formatting to ensure your manuscript meets our submission guidelines. To take advantage of our partnership with AJE, visit the AJE website (https://www.aje.com/go/plos/) for a 15% discount off AJE services. To take advantage of our partnership with Editage, visit the Editage website (www.editage.com) and enter referral code PLOSEDIT for a 15% discount off Editage services. If the PLOS editorial team finds any language issues in text that either AJE or Editage has edited, the service provider will re-edit the text for free. Please note that we will not be able to proceed with publication of your manuscript until the concerns above are addressed. Upon resubmission, please provide the following: * The name of the colleague or the details of the professional service that edited your manuscript* A copy of your manuscript showing your changes by either highlighting them or using track changes (uploaded as a supporting information file)* A clean copy of the edited manuscript (uploaded as the new manuscript file) We look forward to receiving your revised manuscript. Please submit your revised manuscript by Dec 14 2023 11:59PM. If you will need more time than this to complete your revisions, please reply to this message or contact the journal office at plosone@plos.org. Please include the following items when submitting your revised manuscript:

We look forward to receiving your revised manuscript.

Kind regards,

Johanna Pruller, PhD,

Associate Editor, PLOS ONE

on behalf of

Imran Ur Rahman, Ph.D

Academic Editor

PLOS ONE

Journal Requirements:

Additional Editor Comments:

The authors have enhanced the outlook of the article and have addressed previous comments but there are a few minor revisions that need to be addressed. I recommend adding a paragraph explaining Table 5.1 to enhance its clarity for readers. Moreover, I suggest language editing and proof reading to ensure the manuscript is free from grammatical errors. Furthermore, please address the final comments by the reviewers. Their feedback is valuable in enhancing the quality of the manuscript to meet the standards of our publications.

Reviewers' comments:

Reviewer's Responses to Questions

**Comments to the Author**

1. If the authors have adequately addressed your comments raised in a previous round of review and you feel that this manuscript is now acceptable for publication, you may indicate that here to bypass the “Comments to the Author” section, enter your conflict of interest statement in the “Confidential to Editor” section, and submit your "Accept" recommendation.

Reviewer #2: All comments have been addressed

Reviewer #3: All comments have been addressed

2. Is the manuscript technically sound, and do the data support the conclusions?

Reviewer #2: Yes

Reviewer #3: Yes

3. Has the statistical analysis been performed appropriately and rigorously? 

Reviewer #2: Yes

Reviewer #3: Yes

4. Have the authors made all data underlying the findings in their manuscript fully available?

Reviewer #2: Yes

Reviewer #3: Yes

5. Is the manuscript presented in an intelligible fashion and written in standard English?

Reviewer #2: Yes

Reviewer #3: Yes

6. Review Comments to the Author

Reviewer #2: The manuscript is now in good shape. However, authors still need to do a round of language editing because there are still many language improvements needed throughout the manuscript.

Reviewer #3: Dear Authors,

Thank You for the relevant and exciting Article.

Could You please consider the ESG goals achievement in the Discussion section, especially Social dimension in relation with the topic of the research. The Sustainability concept is very important in view of the research.

7. PLOS authors have the option to publish the peer review history of their article (what does this mean?). If published, this will include your full peer review and any attached files.

Reviewer #2: **Yes: **Ajantha Kalyanaratne

Reviewer #3: **Yes: **Sergey Barykin

---

## [Author Response · Author response to Decision Letter 3]

23 Nov 2023

Dear editor, we have submitted the paper to AJE services for revision according to the requirements, and now we are sending you the revised draft, proof of revision and revised draft with identification. Please review it. Thank you very much! Liu Bangfan, 2023-11-14

---

## [Editor Report · Decision Letter 4]

2 Jan 2024

PONE-D-22-34151R4Trade Shocks and Trade Diversion of Epidemic Diseases: Evidence from 110 CountriesPLOS ONE

Dear Dr. Bangfan,

Thank you for submitting your manuscript to PLOS ONE. After careful consideration, we feel that it has satisfied our scientific requirements for publication.

However, our editorial team continues to have significant concerns about the grammar, usage, and overall readability of the manuscript. PLOS ONE requires that published manuscripts use language which is 'clear, correct, and unambiguous', see our criteria for publication at https://journals.plos.org/plosone/s/criteria-for-publication#loc-5. We therefore request that you revise the text to fix the grammatical errors and improve the overall readability of the text. Please note that if the manuscript is not revised sufficiently to meet this publication criterion, it will be rejected.

We suggest you have a fluent English-language speaker thoroughly copyedit your manuscript for language usage, spelling, and grammar. If you do not know anyone who can do this, you may wish to consider employing a professional scientific editing service.

Whilst you may use any professional scientific editing service of your choice, PLOS has partnered with both American Journal Experts (AJE) and Editage to provide discounted services to PLOS authors. Both organizations have experience helping authors meet PLOS guidelines and can provide language editing, translation, manuscript formatting, and figure formatting to ensure your manuscript meets our submission guidelines. To take advantage of our partnership with AJE, visit the AJE website (https://www.aje.com/go/plos/) for a 15% discount off AJE services. To take advantage of our partnership with Editage, visit the Editage website (www.editage.com) and enter referral code PLOSEDIT for a 15% discount off Editage services. If the PLOS editorial team finds any language issues in text that either AJE or Editage has edited, the service provider will re-edit the text for free.

Please note that we will not be able to proceed with publication of your manuscript until the concerns above are addressed.

* A copy of your manuscript showing your changes by either highlighting them or using track changes (uploaded as a supporting information file)

* A clean copy of the edited manuscript (uploaded as the new manuscript file)

We look forward to receiving your revised manuscript.

Kind regards,

Vanessa Carels

Staff Editor

on behalf of

Imran Ur Rahman, Ph.D

Academic Editor

PLOS ONE

Journal Requirements:

Additional Editor Comments (if provided):

All issues have been addressed.
---

## [Author Response · Author response to Decision Letter 4]

3 Feb 2024

Dear editor, we have submitted the paper to AJE services for revision according to the requirements, and now we are sending you the revised draft, proof of revision and revised draft with identification. Please review it. Thank you very much! Liu Bangfan, 2023-11-14

Dear Editor, we have submitted a version of our manuscript on November 14, 2023. This version has been modified by the language professional organization recommended by your magazine. In the process of your review, we were asked to mention it again, but we failed to submit this modified version correctly. We submit this to you again. Please review it. Please forgive me for the trouble I have caused you. Thank you very much for reviewing it again.

Liu Bangfan, 2024-1-5

---

## [Editor Report · Decision Letter 5]

29 Feb 2024

PONE-D-22-34151R5Trade Shocks and Trade Diversion of Epidemic Diseases: Evidence from 110 CountriesPLOS ONE

Dear Dr. Bangfan,

Thank you for submitting your manuscript to PLOS ONE. After careful consideration, we feel that it has merit but does not fully meet PLOS ONE’s publication criteria as it currently stands. Therefore, we invite you to submit a revised version of the manuscript that addresses the points raised during the review process.

For detailed comments, please refer to the "Additional Editor Comments" provided at the end of this email.

We look forward to receiving your revised manuscript.

Kind regards,

Imran Ur Rahman, Ph.D

Academic Editor

PLOS ONE

Journal Requirements:

**Additional Editor Comments:**

Thank you for the revised manuscript. The authors have revised it using AJE services, which has improved the overall quality of the manuscript. The manuscript still has some minor issues that need to be revised:

1) All within-text citations need to be reformatted throughout the manuscript. Citations are reoccurring in two formats, for example (Fernandes et al., 2020) [1] and (Yu Zhen et al., 2020; Dai Chenghao, 2022) [2–3]. I would suggest the authors keep the numbered citations ([1], [2-3]) that follow the guidelines of PLOS ONE and remove the citations with name and year formatting (Yu Zhen et al., 2020; Dai Chenghao, 2022).

2) Please check all the references and ensure they match the citations throughout the manuscript. For example, check reference no. 86.

3) The numbering of headings and sub-headings needs to be updated. For example, after sub-heading 4.3.3 is 4.4.4.

---

## [Author Response · Author response to Decision Letter 5]

2 Mar 2024

Dear Editor: Thank you very much for your revision suggestions!

Three issues raised by the editor about our manuscript have been corrected:

1) All within-text citations need to be reformatted throughout the manuscript. Citations are reoccurring in two formats, for example (Fernandes et al., 2020) [1] and (Yu Zhen et al., 2020; Dai Chenghao, 2022) [2–3]. I would suggest the authors keep the numbered citations ([1], [2-3]) that follow the guidelines of PLOS ONE and remove the citations with name and year formatting (Yu Zhen et al., 2020; Dai Chenghao, 2022).

To solve this problem, we have unified the format according to the PLOS ONE specification.

2) Please check all the references and ensure they match the citations throughout the manuscript. For example, check reference no. 86.

We made one-to-one corrections to the references.

3) The numbering of headings and sub-headings needs to be updated. For example, after sub-heading 4.3.3 is 4.4.4.

We verified and corrected numbering of headings and sub-headings.

Liu Bangfan

2024-03-02

---

## [Editor Report · Decision Letter 6]

24 Mar 2024

Trade Shocks and Trade Diversion of Epidemic Diseases: Evidence from 110 Countries

PONE-D-22-34151R6

Dear Dr. Bangfan,

We’re pleased to inform you that your manuscript has been judged scientifically suitable for publication and will be formally accepted for publication once it meets all outstanding technical requirements.

Kind regards,

Imran Ur Rahman, Ph.D

Academic Editor

PLOS ONE

Additional Editor Comments (optional):

The authors have made the necessary modifications and enhanced the quality of the article. I would suggest the authors make the following changes before proceeding with the publication:

1) Please ensure the denotations of variables within the text follow the proper format. For instance, check the denotation in line 258, 'EDIit', where 'i' and 't' from the model can be added as subscripts.

2) Check and adjust the formatting and uniformaty throughout the paper.
---

## [Editor Report · Acceptance letter]

26 Apr 2024

PONE-D-22-34151R6 

PLOS ONE

Dear Dr. Liu, 

I'm pleased to inform you that your manuscript has been deemed suitable for publication in PLOS ONE. Congratulations! Your manuscript is now being handed over to our production team.

Kind regards, 

on behalf of

Dr. Imran Ur Rahman 

Academic Editor

PLOS ONE